 SHORT REPORT

# VE-cadherin enables trophoblast endovascular invasion and spiral artery remodeling during placental development

Derek C Sung[1], Xiaowen Chen[1], Mei Chen[1], Jisheng Yang[1], Susan Schultz[2], Apoorva Babu[1], Yitian Xu[1], Siqi Gao[1], TC Stevenson Keller IV[1], Patricia Mericko-Ishizuka[1], Michelle Lee[3], Ying Yang[4], Joshua P Scallan[4], Mark L Kahn[1]*

[1]Cardiovascular Institute, Department of Medicine, University of Pennsylvania, Philadelphia, United States; [2]Department of Radiology, Hospital of the University of Pennsylvania, Philadelphia, United States; [3]University Laboratory Animal Resources, University of Pennsylvania, Philadelphia, United States; [4]Department of Molecular Pharmacology and Physiology, University of South Florida, Tampa, United States

*For correspondence:
markkahn@pennmedicine.upenn.edu

**Abstract** During formation of the mammalian placenta, trophoblasts invade the maternal decidua and remodel spiral arteries to bring maternal blood into the placenta. This process, known as endovascular invasion, is thought to involve the adoption of functional characteristics of vascular endothelial cells (ECs) by trophoblasts. The genetic and molecular basis of endovascular invasion remains poorly defined, however, and whether trophoblasts utilize specialized endothelial proteins in an analogous manner to create vascular channels remains untested. Vascular endothelial (VE-) cadherin is a homotypic adhesion protein that is expressed selectively by ECs in which it enables formation of tight vessels and regulation of EC junctions. VE-cadherin is also expressed in invasive trophoblasts and is a prime candidate for a molecular mechanism of endovascular invasion by those cells. Here, we show that VE-cadherin is required for trophoblast migration and endovascular invasion into the maternal decidua in the mouse. VE-cadherin deficiency results in loss of spiral artery remodeling that leads to decreased flow of maternal blood into the placenta, fetal growth restriction, and death. These studies identify a non-endothelial role for VE-cadherin in trophoblasts during placental development and suggest that endothelial proteins may play functionally unique roles in trophoblasts that do not simply mimic those in ECs.

## Editor's evaluation

Understanding molecular and cellular pathways for endovascular invasion and pathogenesis of preeclampsia are important topics for current vascular biology and Ob/Gyn biology, making this study timely and important. It shows for the first time a causal link for the need of VE-cadherin on trophoblasts in vivo for the invasion of these cells into the decidua and for their role in vascular remodeling. The conclusions of this paper are well supported by data. It may provide a novel avenue for the prevention or treatment of preeclampsia.

## Introduction

During placental development in mice and humans, fetal trophoblasts invade the maternal decidua by a process known as endovascular invasion to remodel and connect to maternal spiral arteries (SAs) (*Velicky et al., 2016*; *Soares et al., 2014*; *Hu and Cross, 2011*). This connection allows the flow of

maternal blood through trophoblast-lined sinuses characteristic of hemochorial placentation (*Soares et al., 2018*; *Rai and Cross, 2014*). Shallow trophoblast invasion, deficient SA remodeling, and poor remodeling of the maternal decidua are features of placental dysfunction such as preeclampsia, a hypertensive condition of pregnancy that can lead to maternal and fetal complications (*Lyall et al., 2013*; *Roberts and Escudero, 2012*). Despite the broad and clinically significant impacts of placental dysfunction, the mechanisms controlling trophoblast endovascular invasion and SA remodeling remain poorly defined in vivo.

Invasive trophoblasts are believed to adopt an endothelial-like state by expressing endothelial specific genes (*Nelson et al., 2016*; *Zhou et al., 1997b*; *Govindasamy et al., 2021*), a process that has also been termed 'vascular mimicry' (reviewed in *Rai and Cross, 2014*). Invasive trophoblasts in human and mouse placentas express vascular endothelial (VE)-cadherin (gene name *Cdh5*) during remodeling of SAs (*Zhou et al., 1997b*; *Govindasamy et al., 2021*; *Zhou et al., 1997a*). Invasive trophoblasts in preeclamptic placentas lack VE-cadherin (*Zhou et al., 1997a*), and loss of VE-cadherin reduces trophoblast invasion in vitro (*Cheng et al., 2013*). These studies suggest a functional role for VE-cadherin in endovascular invasion and vessel formation. VE-cadherin is a well-studied cell-cell adhesion protein in the vascular endothelium where it regulates vascular integrity and growth and endothelial barrier function (*Corada et al., 1999*; *Crosby et al., 2005*; *Carmeliet et al., 1999*). In vitro studies have suggested that VE-cadherin may regulate trophoblast-endothelial interactions (*Bulla et al., 2005*), but the requirement for VE-cadherin in trophoblasts during placental development in vivo remains unknown.

In the present study, we functionally tested the role of VE-cadherin in trophoblasts during placental development in mice. We find that conditional deletion of VE-cadherin from trophoblasts disrupts trophoblast invasion into the decidua and SA remodeling, resulting in placental insufficiency and fetal growth restriction. We show that VE-cadherin is important for trophoblasts to interact with and displace SA endothelium. Additionally, trophoblast invasion is important for triggering multiple changes in the decidual extracellular matrix (ECM) and immune cell microenvironment. These studies identify a molecular mechanism by which fetal trophoblasts use VE-cadherin to invade and remodel the maternal environment for successful pregnancy that is relevant to preeclampsia pathogenesis. They also provide a first functional test of the concept of trophoblasts utilizing endothelial programs during endovascular invasion in vivo, and suggest that canonical endothelial proteins may be used by vascular trophoblasts in the placenta in ways that are specific for their function and do not merely mimic endothelial cell (EC) use.

## Results

### Trophoblast-specific deletion of VE-cadherin restricts placental and fetal growth and causes embryonic lethality

To understand the role of VE-cadherin in trophoblasts, we generated *CYP19A1(Tg)^Cre*; *Cdh5^fl/fl* ('*Cdh5* knockout') placentas and mice in which VE-cadherin (encoded by *Cdh5*) is deleted specifically in fetal trophoblasts. Immunostaining for VE-cadherin and the trophoblast marker Cytokeratin 8 (CK8)

**Table 1.** Decreased survival of CYP19A1(Tg)^Cre; Cdh5^fl/fl mutants in late gestation.

Cdh5^fl/fl male mice were crossed with CYP19A1(Tg)^Cre; Cdh5^fl/+ females to generate litters with mixed genotypes. The expected percentage is listed under the genotype label. The observed number of each genotype is shown with the corresponding percentage given in parentheses. P-values were calculated using Fisher's exact test at stages E10.5-12.5 (pooled), E14.5-16.5 (pooled), and P21. E designates embryonic day and P designates postnatal day.

| | CYP19A1(Tg)^Cre; Cdh5^fl/fl **(25%)** | CYP19A1(Tg)^Cre; Cdh5^fl/+ **(25%)** | Cdh5^fl/fl **(25%)** | Cdh5^fl/+ **(25%)** | Total **(100%)** | Fisher Exact Test |
|---|---|---|---|---|---|---|
| E10.5-E12.5 | 13* (26%) | 13 (26%) | 12 (24%) | 12 (24%) | 50 (100%) | *P=1.0000* |
| E14.5-E16.5 | 1 (2.9%) | 13 (38.2%) | 12 (35.3%) | 8 (23.5%) | 34 (100%) | *P=0.0033* |
| P21 | 1 (2.2%) | 11 (24.4%) | 15 (33.3%) | 18 (40.0%) | 45 (100%) | *P=0.0333* |

*One embryo at E12.5 was dead.

demonstrated efficient deletion of VE-cadherin in trophoblasts but not ECs in the *Cdh5* knockout placenta (*Figure 1—figure supplement 1A, B*). *Cdh5* knockout embryos were present at the expected Mendelian ratio at E10.5–12.5 (*Table 1*, p = 1.0000). However, there was an almost complete loss of trophoblast *Cdh5* knockout embryos at E14.5–16.5 and postnatal day 21 (*Table 1*, p < 0.05 and p < 0.005, respectively). Examination of placentas at E12.5 revealed that trophoblast *Cdh5* knockout placentas were smaller and paler than those of control littermates in the same uterus (*Figure 1A–D*). *Cdh5* knockout embryos exhibited marked fetal growth restriction and variable degrees of hemorrhage at E12.5 (green arrowheads, *Figure 1E–H*). No differences in weights were observed between Cre-negative (*Cdh5$^{fl/+}$* or *Cdh5$^{fl/fl}$*) and Cre-positive heterozygous (*CYP19A1(Tg)$^{Cre}$; Cdh5$^{fl/+}$*) placentas and embryos (*Figure 1—figure supplement 1C, D*). Histological and immunofluorescence analysis of E12.5 knockout embryos showed growth defects in numerous organs, including the heart, brain, and liver (*Figure 1—figure supplement 2A-D*). Immunofluorescence staining for VE-cadherin in *CYP19A1(Tg)$^{Cre}$; Cdh5$^{fl/fl}$* embryos show that VE-cadherin is retained in the embryonic vasculature of affected organs, including the brain, heart, liver, lungs, and thorax (*Figure 1—figure supplement 3A-D*). These data demonstrate that trophoblast-specific loss of VE-cadherin confers fetal growth restriction and lethality. Importantly, since *CYP19A1(Tg)$^{Cre}$* activity is present only in placental trophoblasts (*Wenzel and Leone, 2007*; *López-Tello et al., 2019*), these embryonic defects are secondary to placental defects.

## Loss of VE-cadherin disrupts trophoblast endovascular invasion but not formation of the fetal vasculature

To characterize the placental defects conferred by trophoblast loss of VE-cadherin, we performed histological staining with hematoxylin and eosin (H&E) and immunofluorescence staining for CK8 (trophoblasts), Endomucin (endothelial cells), and TER119 (erythrocytes) on serial control and *Cdh5* knockout placenta sections (*Figure 1I–L*). H&E and immunofluorescence staining both showed abundant trophoblasts, marked by CK8 positivity, surrounding Endomucin$^+$ SAs within the decidua in control placentas (red arrowheads, *Figure 1J and J'*). Notably, trophoblasts surrounding maternal SAs were absent in knockout placentas (*Figure 1L and L'*), and quantification of number of trophoblasts and invasion depth into the decidua showed fewer and shallower invasion of trophoblasts overall (decidua, *Figure 1M and N*).

The findings described above suggested that *Cdh5* knockout placentas were less able to carry maternal blood to nourish the growing embryo. Fetal and maternal red blood cells (RBCs) can be differentiated by the presence of nuclei in fetal RBCs. The labyrinth in knockout placentas had fewer enucleated TER119$^+$ RBCs, indicating less maternal blood and consistent with paler placentas observed by gross examination (white arrowheads, *Figure 2J" vs. L"*). To determine whether loss of VE-cadherin affects the fetal placental vasculature, we quantified Endomucin$^+$ vascular area in the labyrinth. We detected no differences in Endomucin$^+$ staining, demonstrating that loss of VE-cadherin from trophoblasts does not disrupt formation of the fetal placental capillary plexus (*Figure 1J, L and O*). These findings suggest that VE-cadherin is important for trophoblast migration and for the association with SAs required to channel maternal blood into the placenta.

## Loss of trophoblast VE-cadherin blocks displacement of SA ECs and SA remodeling

A critical early step in establishing maternal circulation to the placenta is trophoblast invasion of the decidua and its SAs. The observation that there were fewer VE-cadherin-deficient trophoblasts adjacent to SAs and decreased maternal blood within the labyrinth suggested that trophoblast VE-cadherin may play a requisite role in SA remodeling. Since loss of vascular smooth muscle is a key step in SA remodeling, we stained control and knockout placentas for alpha-smooth muscle actin (αSMA). SAs in *Cdh5* knockout placentas exhibited persistent vascular smooth muscle coverage and reduced SA diameter compared to control placentas (white arrowheads, *Figure 2A–C*). These studies reveal that loss of trophoblast VE-cadherin disrupts SA remodeling, which likely contributes to reduced maternal blood within the placenta.

During the process of SA remodeling, invasive trophoblasts displace the endothelial layer of SAs to direct maternal blood flow through trophoblast-lined sinuses into the labyrinth. In vitro studies have suggested that trophoblast expression of VE-cadherin may enable these cells to adhere to SA

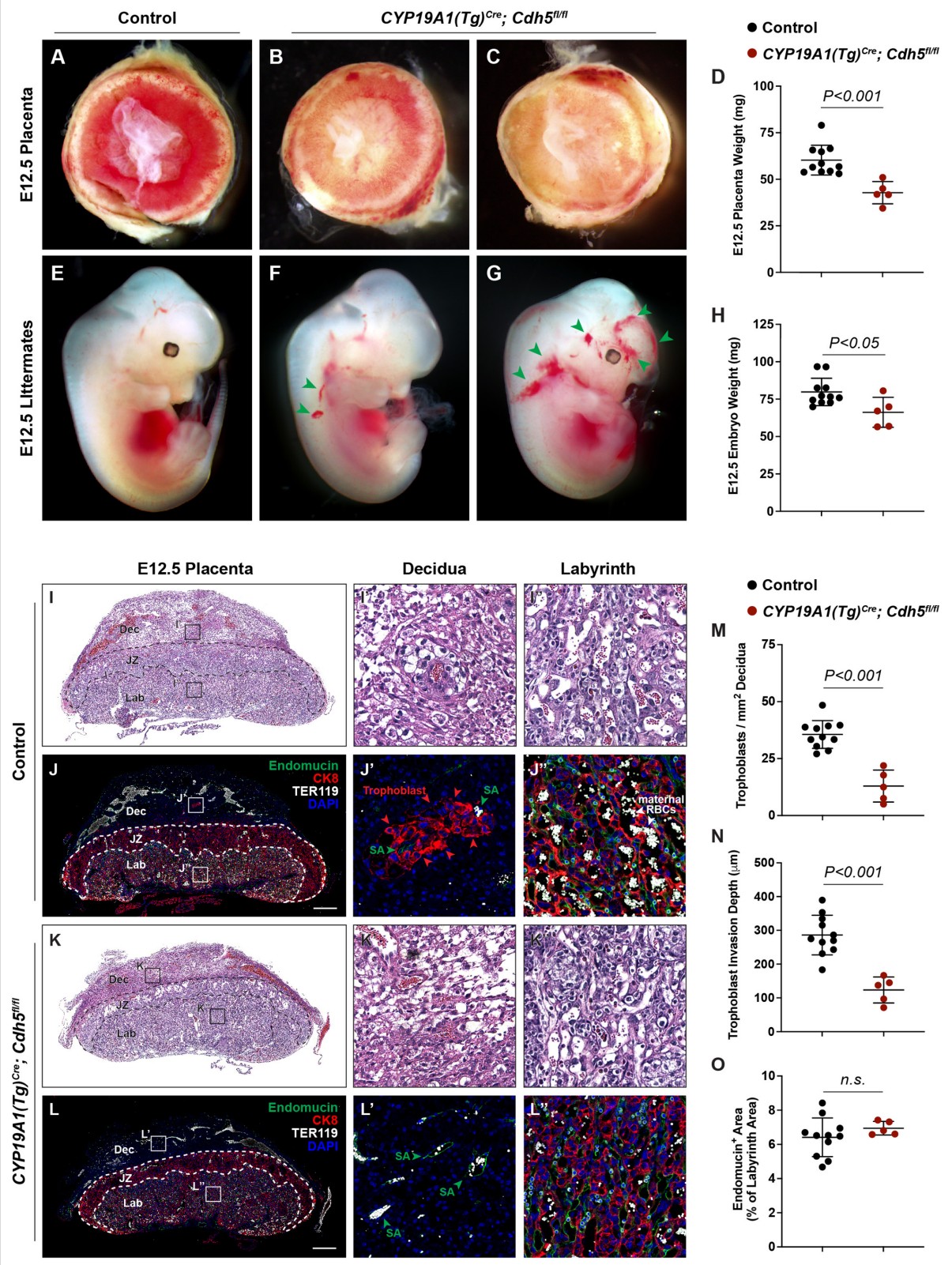

**Figure 1.** Deletion of VE-cadherin in trophoblasts disrupts cell migration resulting in placental and fetal growth restriction. (**A–D**) Gross examination and quantification of E12.5 Control and *CYP19A1(Tg)^Cre^; Cdh5^fl/fl^* placentas and weights. Control n = 11, *CYP19A1(Tg)^Cre^; Cdh5^fl/fl^* n = 5. (**E–H**) Gross examination and quantification of E12.5 Control and *CYP19A1(Tg)^Cre^; Cdh5^fl/fl^* embryos and weights. Control n = 11, *CYP19A1(Tg)^Cre^; Cdh5^fl/fl^* n = 5. Green arrowheads point to areas of hemorrhage. (**I–L**) Hematoxylin and eosin (H&E) staining and immunofluorescence staining for Endomucin (green),

*Figure 1 continued on next page*

*Figure 1 continued*

CK8 (red), and TER119 (gray) of E12.5 Control (**I, J**) and *CYP19A1(Tg)^Cre^; Cdh5^fl/fl^* (**K, L**) serial placenta sections. Dotted lines demarcate the different placental regions. Red arrowheads indicate trophoblasts. Green arrowheads indicate spiral arteries (SA). White arrowheads indicate maternal red blood cells (RBCs). Note fewer non-nucleated, maternal TER119^+ cells in the labyrinth region of *CYP19A1(Tg)^Cre^; Cdh5^fl/fl^* placentas. Boxes on the left correlate with magnified images on the right, and boxes in H&E and immunofluorescence images are of the same region. Scale bars = 500 µm. Dec (decidua), JZ (junctional zone), Lab (labyrinth). (**M–O**) Quantification of number of trophoblasts in the decidua (**M**), trophoblast invasion depth (**N**), and percent labyrinth Endomucin^+ area (**O**). Control n = 11, *CYP19A1(Tg)^Cre^; Cdh5^fl/fl^* n = 5. Statistical analysis was performed using two-tailed, unpaired Welch's t-test. Data are shown as means ± SD.

The online version of this article includes the following source data and figure supplement(s) for figure 1:

**Source data 1.** Excel file containing quantification for embryo weights, placenta weights, trophoblast density, trophoblast migration distance, and fetal labyrinth vasculature in *Figure 1*.

**Figure supplement 1.** Deletion of VE-cadherin in *CYP19A1(Tg)^Cre^; Cdh5^fl/fl^* placentas.

**Figure supplement 1—source data 1.** Excel file containing quantification for VE-cadherin expression in trophoblasts, embryo weights, and placenta weights in *Figure 1—figure supplement 1*.

**Figure supplement 2.** Loss of trophoblast VE-cadherin causes defects in brain, liver, and heart development.

**Figure supplement 2—source data 1.** Excel file containing quantification for liver area and myocardial thickness in *Figure 1—figure supplement 2*.

**Figure supplement 3.** VE-cadherin expression is retained in the vasculature of affected organs in *CYP19A1(Tg)^Cre^; Cdh5^fl/fl^* embryos.

**Figure supplement 3—source data 1.** Excel file containing quantification for VE-cadherin expression in various organs in *Figure 1—figure supplement 3*.

ECs (*Bulla et al., 2005*). The finding that trophoblast *Cdh5* knockout placentas fail to remodel SAs suggested that there may be defects in trophoblast-SA interactions. To address the role of VE-cadherin at the site of trophoblast-SA connection, we first sought to image the site at which trophoblasts connect to SAs. Close inspection of this region in control placentas revealed a clear demarcation from luminal Endomucin^+ SA endothelium to luminal CK8^+ trophoblasts (white arrowheads, *Figure 2D and E*). In contrast, we found that SAs in VE-cadherin knockout placentas maintained a layer of intact ECs despite being surrounded by trophoblasts (white arrowheads, *Figure 2D and E*). Quantification of the percent of trophoblasts immediately in contact with the lumen demonstrated that knockout placentas had a lower percentage of trophoblasts and higher percentage of ECs covering the lumen compared to controls (*Figure 2F*). To better appreciate differences in placental architecture, we additionally performed whole-mount immunofluorescence of thick placental sections. Consistent with our data above, *Cdh5* knockout placentas maintained a layer of ECs in SAs surrounded by trophoblasts and had smaller lumens (*Figure 2G*). Fewer maternal RBCs were seen in trophoblast-lined sinuses in the labyrinth of *Cdh5* knockout placentas, but we found no differences in the fetal capillary plexus (*Figure 2H*). Together, these data suggest that VE-cadherin is required for trophoblast displacement of SA ECs during endovascular invasion for efficient maternal-fetal circulatory connection.

## Defective trophoblast invasion and SA remodeling cause placental insufficiency and fetal distress

The finding that *Cdh5* knockout placentas have less maternal blood and are associated with mid-gestation embryonic lethality suggested that failed SA remodeling restricts maternal blood flow into the placenta, thus causing fetal demise. Human placental insufficiency is typically assessed with ultrasound measurements of placental hemodynamics and fetal heart rate, a readout for overall fetal health. We therefore utilized Doppler ultrasound to measure peak systolic (PSV) and end diastolic velocities (EDV) in the umbilical arteries (*Figure 3A*) and calculated resistance and pulsatility indices (RI and PI) and fetal heart rates to assess placental vascular resistance and fetal wellbeing in individual concepti. Elevated RI and PI values are clinical indicators of placental insufficiency in humans and are associated with conditions such as preeclampsia and fetal growth restriction. Fetal heart rate is used as a clinical parameter for fetal wellbeing, with fetal bradycardia indicative of fetal distress. While control embryos had RIs, PIs, and fetal heart rates within range of previously published values (*Galaz et al., 2020*), we found that trophoblast *Cdh5* knockout embryos exhibited significantly increased RIs and PIs (*Figure 3B–D*) and significantly reduced fetal heart rates (*Figure 3B and E*), consistent with placental insufficiency and fetal distress. Knockout embryos also exhibited reversal of end-diastolic flow as shown by the directional change of velocity from negative (peak systole) to positive (end of

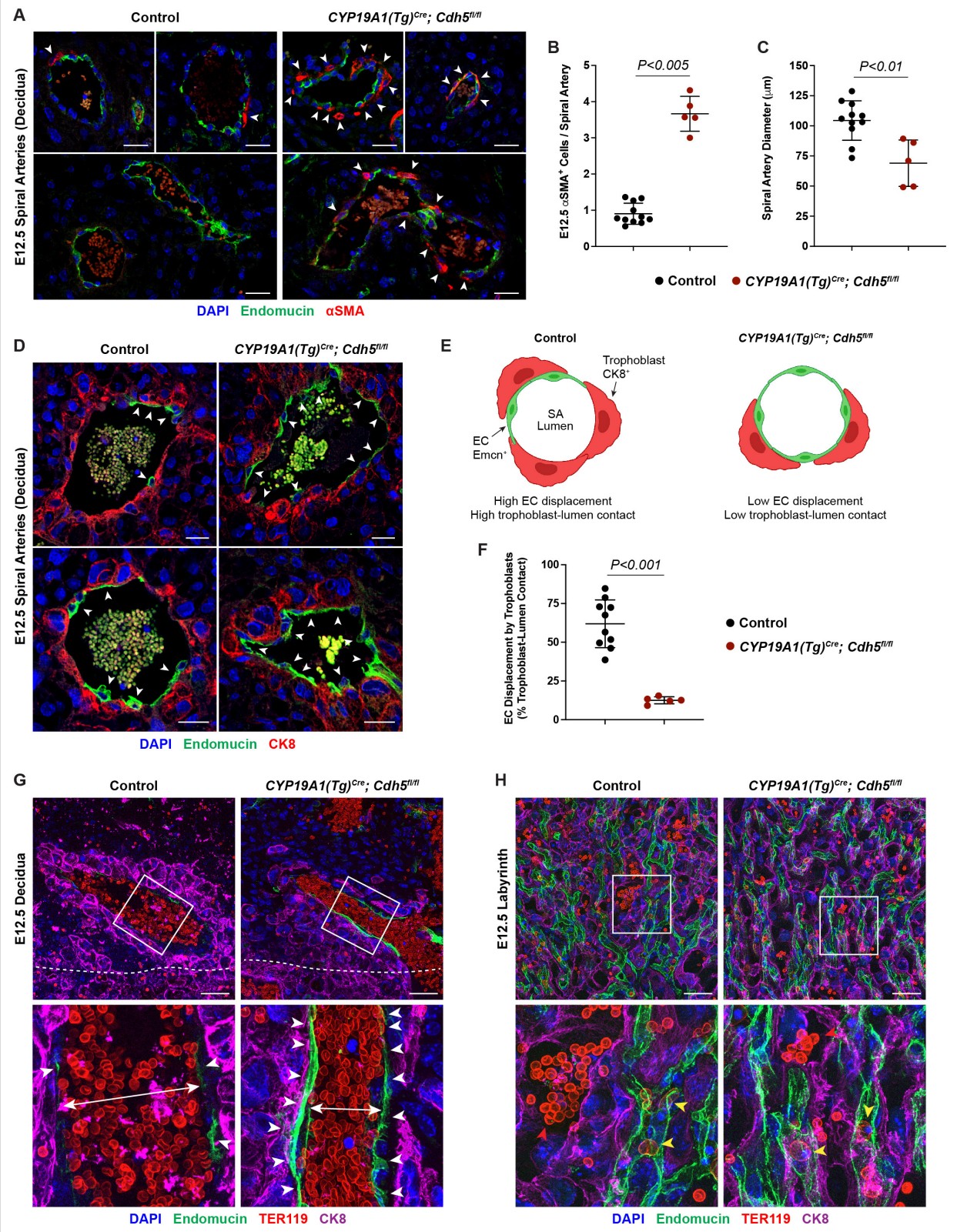

**Figure 2.** VE-cadherin is required in trophoblasts to remodel spiral arteries (SAs) and to displace SA endothelium. (**A**) Immunofluorescence staining of E12.5 Control and *CYP19A1(Tg)^Cre^; Cdh5^fl/fl^* placentas for Endomucin (green) and alpha-smooth muscle actin (αSMA) (red). White arrowheads indicate αSMA⁺ cells. Scale bars = 25 μm. (**B, C**) Quantification of αSMA⁺ cells per SA and SA diameter. Control n = 11, *CYP19A1(Tg)^Cre^; Cdh5^fl/fl^* n = 5. (**D**) Immunofluorescence staining of E12.5 Control and *CYP19A1(Tg)^Cre^; Cdh5^fl/fl^* placentas for Endomucin (green) and CK8 (red). White arrowheads

*Figure 2 continued on next page*

*Figure 2 continued*

indicate Endomucin⁺ SA endothelial cells (ECs). Positive signal in small, rounded cells in the lumen is the result of erythrocyte autofluorescence. (**E**) Schematic demonstrating differences in trophoblast and SA EC contact with the vessel lumen. Scale bars = 25 μm. (**F**) Quantification of the percent trophoblast-lumen contact, which was calculated by measuring the circumference of the vessel lumen and then measuring the length of CK8⁺ trophoblasts in contact with the lumen. Each point represents the average of at least three SAs from an individual placenta. Control n = 10, *CYP19A1(Tg)^Cre^; Cdh5^fl/fl^* n = 5. (**G, H**) Maximum intensity projections of whole-mount immunofluorescence of the decidua (**G**) and labyrinth (**H**) from 200 μm thick placenta sections stained for Endomucin (green), TER119 (red), and CK8 (magenta). Double-headed arrows indicate differences in lumen size. White arrowheads indicate Endomucin⁺ SA ECs. Red arrowheads indicate maternal red blood cells within the trophoblast-lined vessels. Yellow arrowheads indicate fetal red blood cells within fetal capillaries. Dotted white line demarcates the decidua from the junctional zone. Scale bars = 50 μm. Statistical analysis was performed using two-tailed, unpaired Welch's t-test. Data are shown as means ± SD.

The online version of this article includes the following source data for figure 2:

**Source data 1.** Excel file containing quantification for smooth muscle cells per spiral artery, spiral artery diameter, and trophoblast-endothelial cell displacement in *Figure 2*.

diastole) (*Figure 3B*), indicative of high vascular resistance. These hemodynamic data demonstrate placental insufficiency that contributes to fetal growth restriction and fetal demise following loss of trophoblast invasion and SA remodeling.

## Loss of trophoblast VE-cadherin alters decidual ECM remodeling and uterine natural killer cell clearance

Since trophoblast invasion occurs in conjunction with decidual changes, we hypothesized that failed trophoblast migration might affect other placental processes involved in that process. To identify such effects, we performed bulk RNA-sequencing (RNA-seq) on deciduas of E12.5 knockout and control placentas (*Figure 4A*). Analysis of the top 100 differentially expressed genes showed that genes highly expressed in invasive trophoblasts (*Prl4a1, Pla2g4f, Pla2g4d, Nos1, Ncam1, Aldh1a3, Ascl2, Car2, Tfap2c*) (*Nelson et al., 2016*; *Müller et al., 1999*; *Simmons et al., 2008*; *Marsh and Blelloch, 2020*; *Outhwaite et al., 2015*; *Varberg et al., 2021*; *Bogutz et al., 2018*; *Sharma et al., 2016*) were downregulated in knockout placentas (*Figure 4B*), consistent with reduced trophoblasts present in the decidua. Genes previously associated with trophoblast invasion (*Gabrp, Mmp15*) (*Lu et al., 2016*; *Majali-Martinez et al., 2020*) and differentiation (*Cdx2*) (*Ralston and Rossant, 2008*) were also differentially expressed, as were multiple genes related to defective decidualization (*Ccl28, Slc27a2, Klk1, Csf1, Tmem132e, Ermap, Pappa2, Tmc5*) (*Woods et al., 2017*; *Goolam et al., 2020*; *Sun et al., 2013*; *Figure 4B*). Thus RNA-seq data are consistent with defects in trophoblast invasion and suggest the presence of non-cell autonomous effects in the decidua that may contribute to a preeclamptic-like phenotype.

We next analyzed gene ontology (GO) related to cellular components and biological processes. We found that four of the top ten upregulated cellular component GO terms in knockout placentas were related to the ECM (*Figure 4C*). Many of the significantly upregulated genes were secreted ECM proteins (*Vwa2, Ntn1, Fbln1*) (*Figure 4D*). Several extracellular proteases previously linked to abnormal human pregnancies were also differentially expressed. These included increased expression of *Adamts9* (associated with preterm birth; *Li et al., 2020*), reduced *Adamts13* (associated with preeclampsia; *Aref and Goda, 2013*; *Stepanian et al., 2011*), and reduced *Mmp15* (associated with reduced trophoblast invasion; *Majali-Martinez et al., 2020*). GO analysis of biological processes revealed that seven of the top ten upregulated GO terms were related to the immune system and immune activation, specifically the innate immune response (*Figure 4E*). Many of the significantly upregulated genes were involved in the complement factor pathway (*C3, Cfb, Cfh*) (*Figure 4F*). Thus GO analysis suggests that defects in ECM remodeling and changes in immune cells may contribute to the placental defects associated with reduced trophoblast invasion of the decidua.

We next aimed to directly assess the impact of trophoblast-specific deletion of VE-cadherin on (1) ECM remodeling and (2) immune cells. Our RNA-seq data showed a >2.5-fold decrease in *Mmp15* (*Figure 4D* and *Figure 4—figure supplement 1A*). MMP15 (also known as MT2-MMP) is a membrane metalloprotease that is expressed in invasive trophoblasts in first trimester human placentas that promotes trophoblast invasion and degrades laminin as gestation progresses (*Majali-Martinez et al., 2020*; *Pollheimer et al., 2014*; *Majali-Martinez et al., 2016*; *Turpeenniemi-Hujanen et al., 1995*). Examination of MMP15 expression and its target laminin using immunofluorescence revealed increased

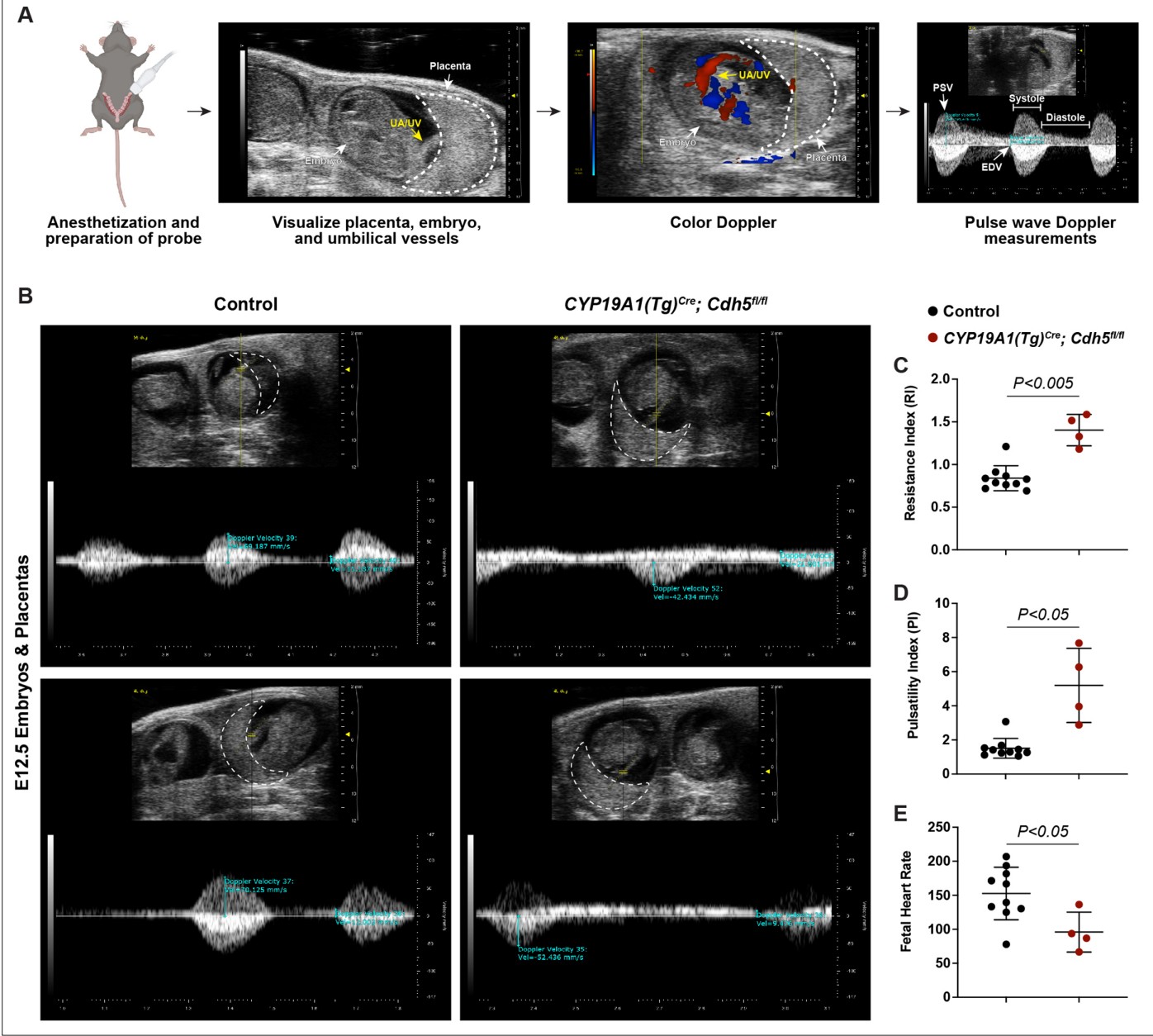

**Figure 3.** Spiral artery remodeling defects result in placental insufficiency and fetal distress. (**A**) Schematic of workflow for Doppler ultrasound of pregnant dams. The embryo, UA/UV (yellow arrow), and placenta (dotted white outline) are labeled. (**B**) Representative umbilical artery Doppler waveforms from two Control and two *CYP19A1(Tg)^Cre; Cdh5^fl/fl* placentas. The placenta is outlined in a dotted white line. Reversal of end-diastolic flow is evident by the change of directional velocity at the end of diastole compared to peak systole (i.e., negative to positive velocity). (**C–E**) Quantification of resistance index (RI), pulsatility index (PI), and fetal heart rate. PSV (peak systolic velocity), EDV (end diastolic velocity), UA/UV (umbilical artery/umbilical vein). Note that red/blue colors in color Doppler images do not indicate UA/UV, which can only be differentiated based on the Doppler waveform. Control n = 10, *CYP19A1(Tg)^Cre; Cdh5^fl/fl* n = 4. Statistical analysis was performed using two-tailed, unpaired Welch's t-test. Data are shown as means ± SD.

The online version of this article includes the following source data for figure 3:

**Source data 1.** Excel file containing quantification for ultrasound studies (resistance index, pulsatility index, fetal heart rate) in *Figure 3*.

laminin in the decidual stroma of *Cdh5* knockout placentas but no differences in MMP15 expression in trophoblasts (*Figure 4—figure supplement 1B, C*). Additionally, we evaluated vinculin, a focal adhesion protein that regulates cell-matrix adhesion and associates with VE-cadherin (*Huveneers et al., 2012*). Vinculin is required for cell polarization and invasion (*Thievessen et al., 2015*; *Mierke*

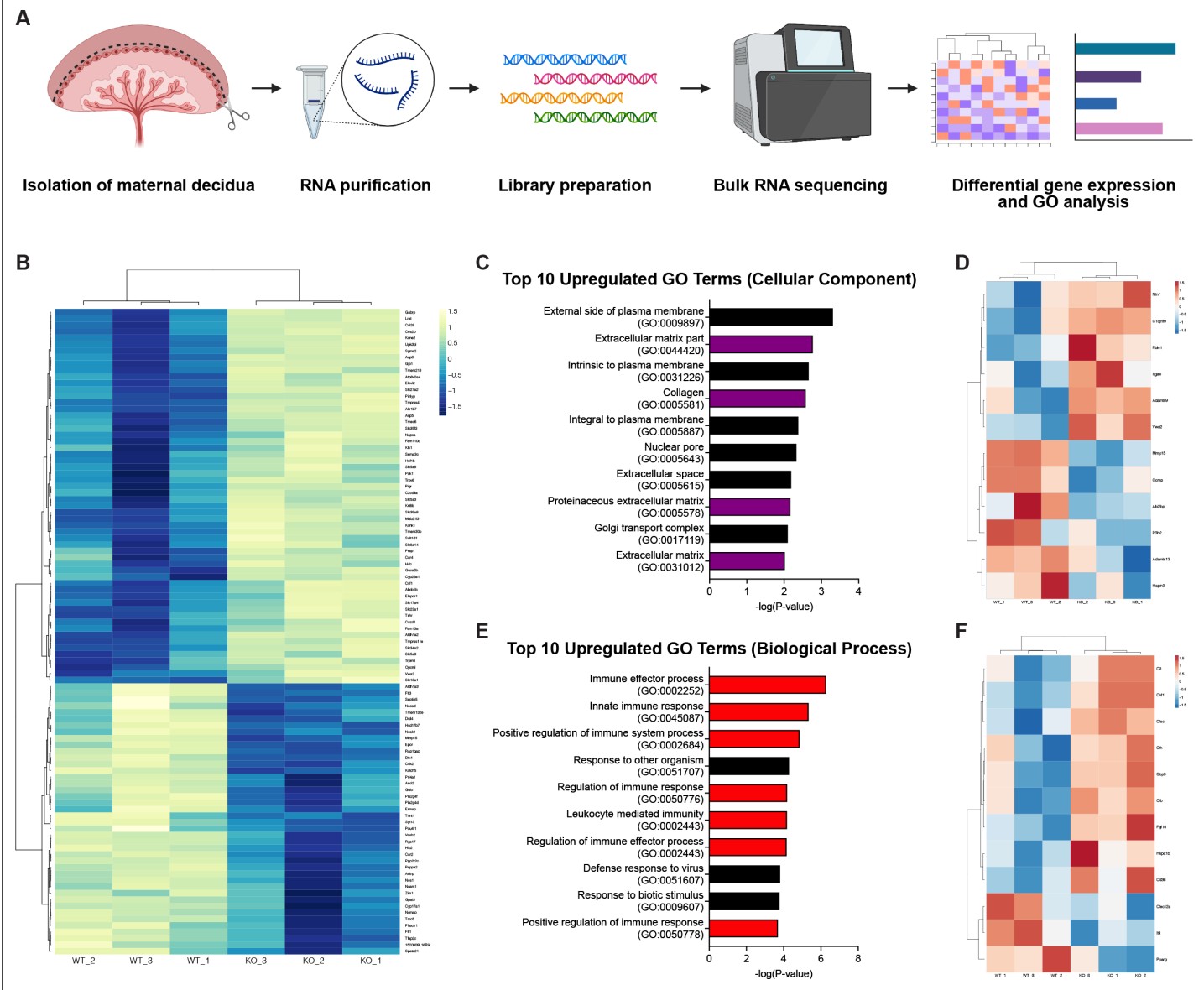

**Figure 4.** RNA-sequencing reveals defects in the decidual extracellular matrix and immune microenvironment. (**A**) Schematic of bulk RNA-sequencing (RNA-seq) workflow on deciduas from Control (WT) and *CYP19A1(Tg)^Cre*; *Cdh5^fl/fl* (KO) E12.5 placentas. (**B**) Heatmap of the top 100 differentially regulated genes shown by z-score (n = 3 biological replicates). (**C**) Gene ontology (GO) term analysis of top 10 upregulated cellular components in *CYP19A1(Tg)^Cre*; *Cdh5^fl/fl* placentas. Purple bars indicate GO terms related to the extracellular matrix. (**D**) Heatmap of significantly differentially expressed genes from GO terms related to the extracellular matrix. (**E**) GO term analysis of top 10 upregulate biological processes in *CYP19A1(Tg)^Cre*; *Cdh5^fl/fl* placentas. Red bars indicate GO terms related to immune processes. (**F**) Heatmap of significantly differentially expressed genes from GO terms related to immune processes.

The online version of this article includes the following source data and figure supplement(s) for figure 4:

**Figure supplement 1.** Disrupting trophoblast migration results in decidual extracellular matrix defects.

**Figure supplement 1—source data 1.** Excel file containing quantification for Mmp15 gene expression, laminin expression in the decidua, and vinculin expression in trophoblasts in *Figure 4—figure supplement 1*.

**Figure supplement 2.** Persistent uterine natural killer (uNK) cells at the junctional zone-decidual interface of *Cdh5* knockout placentas.

**Figure supplement 2—source data 1.** Excel file containing quantification for uterine natural killer cell density and apoptotic cell density in *Figure 4—figure supplement 2*.

*et al., 2010*), and vinculin levels were indeed decreased in invasive trophoblasts of *Cdh5* knockout placentas (*Figure 4—figure supplement 1D*). Together, these results suggest that VE-cadherin cell autonomously controls focal adhesions but not MMP activity and that persistent laminin in the decidua of *Cdh5* knockout placentas is likely a consequence of fewer MMP15-expressing trophoblasts.

Since uterine natural killer (uNK) cells are the most abundant innate immune cell in the decidua, we stained placenta sections with the uNK cell-specific marker DBA. We found markedly increased uNK cells in knockout placentas compared to controls (*Figure 4—figure supplement 2A, B*). In order to determine the cause of increased uNK cells, we stained for the apoptosis marker cleaved caspase-3 and found numerous apoptotic uNK cells in control placentas; however almost no uNK cells in knockout placentas were positive for cleaved caspase-3 (*Figure 4—figure supplement 2A, C*). Lastly, we also found multiple significantly upregulated NK cell-related GO terms reflecting increased NK cell activity (*Figure 4—figure supplement 2D*). Together, these results indicate that loss of trophoblast invasion has non-cell autonomous effects that impact that maternal microenvironment of the placenta.

## Discussion

The invasion of the maternal decidua by trophoblasts and their fusion to maternal SAs is a critical step in establishing placental circulation. However, the mechanisms by which trophoblast migration and endovascular invasion are accomplished remain largely unknown. Trophoblasts express endothelial molecular and genetic programs during invasion of SAs (*Soares et al., 2018*). However, the function of specific endothelial genes in trophoblasts has not been functionally assessed in vivo, and this model remains untested. In blood and lymphatic ECs, VE-cadherin is used to maintain vascular integrity (*Corada et al., 1999*; *Crosby et al., 2005*; *Carmeliet et al., 1999*), restrict endothelial migration (*Crosby et al., 2005*; *Hägerling et al., 2018*), and regulate angiogenic growth (*Gaengel et al., 2012*). While it is an attractive concept that trophoblasts may form vascular sinuses using similar genetic programs, the findings that loss of VE-cadherin decreases trophoblast cell migration and prevents SA remodeling suggest that trophoblasts utilize VE-cadherin in a manner distinct from ECs. Our work characterizing mechanisms of endovascular invasion in the placenta suggests that the use of endo-thelial proteins by trophoblasts may be relatively specific to their role in the placenta and not a simple reflection of vascular EC function.

VE-cadherin is primarily an adhesive receptor that acts in a homotypic manner to establish strong EC-cell junctions. Evidently, VE-cadherin is required in trophoblasts to invade the maternal decidua and remodel the maternal microenvironment. Our findings that loss of VE-cadherin decreases vinculin but not MMP15 in trophoblasts suggest that VE-cadherin mainly regulates cell invasion and that ECM remodeling defects are likely secondary consequences of decreased trophoblast invasion. A second interesting aspect of *Cdh5* knockout placentas is persistent innate immune cells within the decidua. uNK cells (called decidual NK cells or dNK cells in humans) are present in the mouse decidua at E6.5, prior to formation and invasion of trophoblasts, and decline in number beginning at E12.5 (*Rajagopalan, 2014*; *Sojka et al., 2018*). uNK cells also secrete factors such as VEGF-C that promote SA remodeling (*Pawlak et al., 2019*). Significantly, increased dNK cells is also characteristic of preeclamptic placentas (*Zhang et al., 2019*), similar to our mouse model. Our findings raise the possibility that trophoblast migration into the decidua may coordinate decidual matrix and immune changes that promote SA remodeling.

In humans, defective SA remodeling and shallow trophoblast invasion are hallmarks of preeclampsia. Preeclampsia is a complex and heterogeneous disease with maternal and fetal contributions to its pathogenesis, and many in vitro models fail to fully recapitulate many aspects of its pathophysi-ology. Most rodent models of preeclampsia utilize maternal genetic or pharmacological perturba-tions (*Gatford et al., 2020*; *Marshall et al., 2018*), and there have been few in vivo models in which preeclamptic features are recapitulated with fetal modulation of trophoblasts. Previous studies of human placentas showed that invasive trophoblasts in severely preeclamptic placentas exhibit reduced expression of VE-cadherin (*Zhou et al., 1997a*) however whether this is a cause or consequence of placental dysfunction has been unclear. Our mouse model utilizing trophoblast-specific knockout of VE-cadherin exhibits many histopathological and clinical features of preeclampsia and suggests that loss of VE-cadherin in trophoblasts may be a primary contributor to preeclampsia pathogenesis. Addi-tionally, we observe secondary defects in organogenesis and vascular development in the embryo (*Perez-Garcia et al., 2018*), which has also been linked to placentation defects. Interestingly, VEGF-A

is known to induce VE-cadherin expression in cultured trophoblasts (*Chang et al., 2005*) and may therefore be a useful strategy in treating preeclampsia. Trophoblast-specific loss of VE-cadherin may serve as a useful model for studying fetal contributions to preeclampsia.

## Materials and methods

**Key resources table**

| Reagent type (species) or resource | Designation | Source or reference | Identifiers | Additional information |
|---|---|---|---|---|
| Genetic reagent (*Mus musculcus*) | CYP19A1(Tg)-Cre | *Wenzel and Leone, 2007* | | |
| Genetic reagent (*Mus musculcus*) | *Cdh5* flox | *Yang et al., 2019* | | |
| Antibody | Anti-Endomucin (goat polyclonal) | R&D | AF4666 | IF(1:400) |
| Antibody | Anti-Endomucin (rat monoclonal) | Abcam | ab106100 | IF(1:300) |
| Antibody | Anti-TER119 (rat monoclonal) | Abcam | ab91113 | IF(1:300) |
| Antibody | Anti-VE-cadherin (goat polycloncal) | R&D | AF1002 | IF(1:200) |
| Antibody | Anti-CK8 (rabbit monoclonal) | Abcam | ab53280 | IF(1:300) |
| Antibody | Anti-CK8 (rat monoclonal) | DSHB | TROMA-1 | IF(1:400) |
| Antibody | Anti-αSMA-Cy3 (mouse monoclonal) | Sigma | C6198 | IF(1:300) |
| Antibody | Anti-Cleaved Caspase-3 (rabbit polyclonal) | Millipore Sigma | AB3623 | IF(1:100) |
| Antibody | Anti-MMP15 (rabbit polyclonal) | Thermo Fisher Scientific | PA5-13184 | IF(1:200) |
| Antibody | Anti-Laminin (rabbit polyclonal) | Sigma | L9393 | IF(1:200) |
| Antibody | Anti-Vimentin (goat polyclonal) | R&D | AF2105 | IF(1:300) |
| Antibody | Anti-Vinculin (mouse monoclonal) | Sigma | V9131 | IF(1:200) |
| Commercial assay or kit | Direct-zol RNA Miniprep Kits | Zymo Research | R2053 | |
| Software, algorithm | ImageJ | NIH, Bethesda, MD, USA | RRID:SCR_003070 | |
| Software, algorithm | GraphPad Prism | GraphPad | RRID:SCR_002798 | |
| Software, algorithm | Picard v2.17.11 | Picard | RRID:SCR_006525 | |
| Other | DBA-Biotin | Vector Labs | B-1035 | (1:500) |

### Generation of mutant mice

*CYP19A1(Tg)-Cre* mice have been previously described (*Wenzel and Leone, 2007*) in which the transgene relies on Cre expression under a 501 bp region with the first exon of human *CYP19A1* containing regulatory elements for trophoblast-specific expression, as *CYP19A1* is not endogenously expressed in trophoblasts. VE-cadherin (*Cdh5*) floxed mice have been previously described (*Yang et al., 2019*) and were generated with LoxP sites flanking exons 3 and 4. Mice were bred according to standard protocols and maintained on a mixed background. Male *Cdh5*[fl/fl] mice were mated to female *CYP19A1(Tg)*[Cre]; *Cdh5*[fl/+] mice due to the influence of parental inheritance on Cre expression, with maternal inheritance providing the most robust and consistent expression (*Wenzel and Leone, 2007*). Mating pairs were set up in the afternoon and vaginal plugs checked in the morning. Presence of a vaginal plug indicated embryonic day (E)0.5. Cre-negative (*Cdh5*[fl/+] and *Cdh5*[fl/fl]) and Cre-positive heterozygous (*CYP19A1(Tg)*[Cre]; *Cdh5*[fl/+]) littermates were used as controls. All procedures were conducted using an approved animal protocol (806811) in accordance with the University of Pennsylvania Institutional Animal Care and Use Committee.

### Intrauterine Doppler ultrasound

In utero Doppler ultrasound was performed by a trained technician using the VEVO2100 Ultrasound System equipped with the MS-400 transducer (30 MHz). E12.5 pregnant mice were lightly anesthetized using 2% isoflurane. Hair was removed from the abdomen using chemical hair remover (Nair),

and the animals were placed on a warming pad. Maternal heart rate and temperature were continuously monitored and consistently within 400–500 bpm and 37°C. Ultrasound gel was applied to the abdomen and the transducer applied to visualize embryos and placentas using the maternal bladder as an anatomical landmark. Color Doppler was used to visualize the umbilical vessels, and pulse wave (angle of insonation <60°) measurements were made at the point where the umbilical artery inserts into the placenta. From the Doppler waveforms, PSV and EDV were measured and used to calculate resistance index [$RI = 1 - PSV/EDV$] and pulsatility index [$PI = (PSV - EDV)/mean\ velocity$, where mean velocity = $(PSV + EDV)/2$]. Heart rate was calculated as beats per minute by dividing 60 s by the systolic + diastolic time.

## Histology and immunofluorescence staining and analysis

Whole mouse embryos or placentas were collected and fixed in 4% paraformaldehyde (PFA) overnight at 4°C prior to dehydration in alcohol and paraffin embedding. Tissue sections underwent to dewaxing and rehydration through xylene and ethanol treatment and were then subject to H&E staining or processed for immunofluorescence. For immunodetection, 10 mM citrate buffer (pH 6) was used for antigen retrieval, and sections were blocked with 10% donkey serum in 1% BSA prior to primary antibody treatment overnight at 4°C. A list of antibodies can be found below. Fluorescence-conjugated Alexa Fluor secondary antibodies were used (1:500, Invitrogen) according to the primary antibody species and counterstained with DAPI (1:1000). Sections or tissues were mounted on slides with ProLong Gold Antifade reagent. Signals were detected and images collected using a Zeiss LSM 880 confocal microscope and Zeiss Axio Observer 7 widefield microscope. Images were visualized using ImageJ/FIJI software (NIH).

## Whole-mount immunofluorescence

Whole mouse placentas were collected and fixed in 4% PFA overnight at 4°C and then placed in 1× PBS. Placentas were embedded in 3% low-melt agarose and cut into 200 µm thick sections using a vibratome. Sections were permeabilized with 0.2% Triton X-100 and blocked with 10% donkey serum in 1% BSA prior to primary antibody treatment overnight at 4°C. Fluorescence-conjugated Alexa Fluor secondary antibodies were used (1:500, Invitrogen) according to the primary antibody species and counterstained with DAPI (1:1000). Sections were mounted on a glass slide in a silicone isolator and filled with ProLong Gold Antifade reagent. Signals were detected and images collected using a Zeiss Axio Observer 7 widefield microscope with the Apotome 3 attachment for optical sectioning. Raw images were deconvoluted using ZEN Blue. Maximum intensity projections were generated using ImageJ/FIJI software (NIH).

## Quantification of immunofluorescence images

Number of CK8$^+$ trophoblasts were manually counted and divided by total decidual area. Trophoblast invasion depth was measured as the distance the farthest CK8$^+$ trophoblast was found in the decidua relative to the junctional zone. Percent labyrinth EC coverage was measured by quantifying Endomucin-positive area as a percentage of total labyrinth area. Number of αSMA$^+$ smooth muscle cells were manually counted and divided by number of SAs. SA diameter was manually measured using only circular SAs, taking the average of the long- and short-axis diameters, and averaging at least five SAs per section per placenta. Trophoblast-endothelial displacement was quantified by measuring the circumference of the vessel lumen and then measuring the length of CK8$^+$ trophoblasts in contact with the lumen. Percent displacement was calculated by dividing the trophoblast length by the lumen circumference. VE-cadherin deletion was quantified by measuring VE-cadherin$^+$CK8$^+$ area (determined by Colocalization Threshold plugin in FIJI) and calculating it as a percent of CK8$^+$ area. The Colocalization Threshold plugin was also used to generate colocalization images in *Figure 1—figure supplement 3C*. Laminin and vinculin mean fluorescence intensity was calculated by dividing total fluorescence intensity by area. All images were analyzed using ImageJ/FIJI software.

## Bulk RNA-seq

Total RNA was extracted from E12.5 deciduas (n = 3 per genotype) using Direct-zol RNA miniprep kit (Zymo Research). Quality assessment, cDNA library synthesis, and sequencing using Illumina HiSeq with a 2 × 150 configuration were conducted through GeneWiz. Fastq files were assessed for quality

control using the FastQC program (v0.11.7). They were then aligned against the mouse reference genome (mm39) using the STAR aligner (v2.7.8a) (*Dobin et al., 2013*). Duplicate reads were flagged using the MarkDuplicates program from Picard tools (v2.17.11) (http://broadinstitute.github.io/picard/). Per gene read counts for Ensembl (GRCm39) gene annotations were computed using the R package Rsubread (*Liao et al., 2019*) and duplicate reads were removed. Gene counts were normalized as counts per million (CPM) using the R package edgeR (*Robinson et al., 2010*) and genes with CPM < 1 in 25% of samples were filtered out. The data was transformed using the VOOM function from the limma R package (*Law et al., 2014*). Differential gene expression was performed as a paired analysis using limma. p-Values were adjusted for multiple comparisons using Benjamini-Hochberg procedure. Genes with adjusted p-values less than 0.05 and an absolute $\log_2$-fold change >1 were considered significantly differentially expressed genes. The RNA-seq data set has been deposited in the NCBI GEO under accession ID number GSE189408.

## Statistical analysis

All data are reported as means with n ≥ 3 independent experiments or mice, and error bars represent standard deviation. Each data point in the figures represents one individual placental or embryo. The explicit number of samples is indicated in the figure legends. No explicit power analyses were used to predetermine sample size, and no randomization was used. No samples were excluded for analysis. Statistical significance was determined using Welch's t-test. Differences between means were considered significant at p < 0.05. Significant differences in expected genotypes was calculated using two-tailed Fisher's exact test and considered statistically significant at p < 0.05.

## Acknowledgements

We thank members of the Kahn laboratory and Mainigi laboratory for thoughtful discussions during the course of these studies, the CDB Microscopy Core for support with microscopy, and the Small Animal Imaging Facility Core for their support with ultrasound imaging. We thank Dr Jeremy Veenstra-Vanderweele (Columbia University) and Dr Gustsavo Leone (Medical University of South Carolina) for kindly providing the *CYP19A1(Tg)*[Cre] mice. This work was supported by NIH grants T32 HL007439 and F30 HL158014 (to DCS), American Heart Association Postdoctoral Fellowship No. 35200213 (to XC), NIH grant T32 HL007971 and American Heart Association Postdoctoral Fellowship No. 836,238 (to TCSK), NIH R01 grant HL142905 (to JPS), NIH R01 grant HL145397 (to YY), and NIH R01 grant HL142976 (to MLK).

## Additional information

### Funding

| Funder | Grant reference number | Author |
| --- | --- | --- |
| National Institutes of Health | T32 HL007439 | Derek C Sung |
| National Institutes of Health | F30 HL158014 | Derek C Sung |
| American Heart Association | Postdoctoral Fellowship 35200213 | Xiaowen Chen |
| National Institutes of Health | T32 HL007971 | Thomas C Stevenson Keller |
| American Heart Association | Postdoctoral fellowship 836238 | Thomas C Stevenson Keller |
| National Institutes of Health | HL142905 | Joshua P Scallan |
| National Institutes of Health | HL145397 | Ying Yang |

| Funder | Grant reference number | Author |
|---|---|---|
| National Institutes of Health | HL142976 | Mark L Kahn |

The funders had no role in study design, data collection and interpretation, or the decision to submit the work for publication.

## Author contributions

Derek C Sung, Conceptualization, Investigation, Methodology, Writing - original draft, Writing - review and editing; Xiaowen Chen, Investigation, Validation; Mei Chen, Susan Schultz, Investigation, Methodology; Jisheng Yang, Yitian Xu, TC Stevenson Keller, Investigation; Apoorva Babu, Data curation, Software; Siqi Gao, Formal analysis, Investigation, Methodology; Patricia Mericko-Ishizuka, Methodology, Project administration; Michelle Lee, Methodology; Ying Yang, Joshua P Scallan, Resources; Mark L Kahn, Conceptualization, Investigation, Methodology, Supervision, Validation, Writing - original draft, Writing - review and editing

## Author ORCIDs

Derek C Sung (ID) http://orcid.org/0000-0001-6966-2596
Mark L Kahn (ID) http://orcid.org/0000-0002-6489-7086

## Ethics

All procedures were conducted using an approved animal protocol (806811) in accordance with the University of Pennsylvania Institutional Animal Care and Use Committee.

## Decision letter and Author response

Decision letter https://doi.org/10.7554/eLife.77241.sa1
Author response https://doi.org/10.7554/eLife.77241.sa2

# Additional files

## Supplementary files

• Transparent reporting form

## Data availability

Source Data files have been included for Figure 1, Figure 1—figure supplement 1, Figure 1—figure supplement 2, Figure 1—figure supplement 3, Figure 2, Figure 3, Figure 4—figure supplement 1, and Figure 4—figure supplement 2. All reagents have been listed in the Methods section in this paper. The RNA-seq data set has been deposited in the NCBI GEO under accession ID number GSE189408. Investigators interested in the animals used in this study should contact Dr. Jeremy Veenstra-Vanderweele (Columbia University), Dr. Gustsavo Leone (Medical University of South Carolina), and Dr. Joshua Scallan (University of South Florida).

The following dataset was generated:

| Author(s) | Year | Dataset title | Dataset URL | Database and Identifier |
|---|---|---|---|---|
| Derek C S, Mark L K, Apoorva B | 2021 | VE-cadherin is essential for trophoblast migration and endovascular invasion | http://www.ncbi.nlm.nih.gov/geo/query/acc.cgi?acc=GSE189408 | NCBI Gene Expression Omnibus, GSE189408 |

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
