## [Editor Report]

Understanding molecular and cellular pathways for endovascular invasion and pathogenesis of preeclampsia are important topics for current vascular biology and Ob/Gyn biology, making this study timely and important. It shows for the first time a causal link for the need of VE-cadherin on trophoblasts in vivo for the invasion of these cells into the decidua and for their role in vascular remodeling. The conclusions of this paper are well supported by data. It may provide a novel avenue for the prevention or treatment of preeclampsia.

---

## [Decision Letter]

**Decision letter after peer review:**

Thank you for submitting your article "Vascular mimicry by VE-cadherin enables trophoblast endovascular invasion and spiral artery remodeling during placental development" for consideration by *eLife*. Your article has been reviewed by 3 peer reviewers, including Gou Young Koh as Reviewing Editor and Reviewer #3, and the evaluation has been overseen by Ricardo Azziz as the Senior Editor. The following individual involved in review of your submission has agreed to reveal their identity: Lena Claesson-Welsh (Reviewer #1);

The comments of all three reviewers are in good agreement. While the reviewers found this study is well-designed and well-presented, high quality, and to be of importance as a translational research, they raised concerns about the lack of mechanistic insights to support the authors' claims. The authors are required to carefully address the comments point-by-point in a data-driven manner or with further analyses and discussions. Specifically, the authors are encouraged to pay attention to the major comment 1 of reviewer 1, comments 1 and 3 of reviewer 2, and comment 3 of reviewer 3.

*Reviewer #1 (Recommendations for the authors):*

1. I would like to question the use of the term "Vascular mimicry" in the context of trophoblast invasion and spiral artery remodelling since it is so strongly associated with the formation of non-endothelial channels in cancer, independently of normal blood vessels or angiogenesis. VE-cadherin is clearly not uniquely expressed in blood vascular endothelial cells. Moreover, as stated by the authors, the role of VE-cadherin in trophoblasts appears to be quite different from that in blood vascular endothelial cells. Perhaps endothelial mimicry is a better term, alternatively, just use "endovascular invasion".

2. Deletion of Cdh5 from trophoblasts is shown in Figure supplement 1 but it would be important to also have the data quantified.

3. The authors state that Cre-negative (Cdh5fl/+ and Cdh5fl/fl) and Cre-positive heterozygous (Cyp19Cre; Cdh5fl/+) littermates were used as controls. Since Cre toxicity is a well-known problem, it should be shown which type of control mice were used in which experiments.

*Reviewer #2 (Recommendations for the authors):*

1) It should be demonstrated that Cyp19-Cre driven gene inactivation of VE-cadherin does not affect VE-cadherin expression in endothelial cells of the embryo in tissues where bleeding was observed.

2) The results shown in figures 2 and 3 should be confirmed by whole mount staining of either vibratome sections or cleared tissue.

3) It should be attempted to get some mechanistic insights by analyzing the effects of VE-cadherin silencing on the expression of genes that could be relevant for cell migration, collective migration or association of trophoblast cell clusters.

4) The discussion is rather short and does not attempt to debate mechanistic aspects of the findings.

*Reviewer #3 (Recommendations for the authors):*

The questions and suggestions below are therefore not intended to trigger significant additional experimental work for the present paper, but to perhaps increase clarity and make the study more accessible to readers.

1. Please explain why the Cdh5 knockout embryos exhibited marked hemorrhage at E12.5 (Figure 1E).

2. What about the diameter of the spiral artery in the Cdh5 knockout placentas versus WT placentas?

3. It would be nice if the authors performed scRNA-seq of the trophoblasts of WT and Cdh5 knockout placentas and compared them to provide molecular insight into how the lack of Cdh5 in the trophoblasts led to such abnormalities.

4. Please describe any plausible ways to normalize Cdh5 level in the trophoblasts for enhanced endovascular invasion, which can be an alternative way for prevention of preeclampsia.

---

## [Author Response]

Reviewer #1 (Recommendations for the authors):1. I would like to question the use of the term "Vascular mimicry" in the context of trophoblast invasion and spiral artery remodelling since it is so strongly associated with the formation of non-endothelial channels in cancer, independently of normal blood vessels or angiogenesis. VE-cadherin is clearly not uniquely expressed in blood vascular endothelial cells. Moreover, as stated by the authors, the role of VE-cadherin in trophoblasts appears to be quite different from that in blood vascular endothelial cells. Perhaps endothelial mimicry is a better term, alternatively, just use "endovascular invasion".

We agree that the term “vascular mimicry” is not appropriate given its use in cancer biology and non-endothelial functions of trophoblasts. To avoid confusion, we have removed it from the title of our manuscript (which now reads as “VE-cadherin enables trophoblast endovascular invasion and spiral artery remodeling during placental development”) and from our revised discussion.

2. Deletion of Cdh5 from trophoblasts is shown in Figure supplement 1 but it would be important to also have the data quantified.

In the revised manuscript we provide new data quantifying VE-cadherin deletion in trophoblasts. Briefly, we utilized the Colocalization Threshold plugin in FIJI to determine area of VE-cadherin^+^CK8^+^ positivity in the decidua. This area was divided by the total CK8^+^ area and taken as a percentage. We found an approximately 80% decrease in VE-cadherin expression in trophoblasts in knockout placentas vs. controls. These data are now included as Figure 1—figure supplement 1A, B and the detailed in the Methods section. As noted in our response to Reviewer 3’s comments, the alternative approach of measuring CDH5 using flow cytometry of trophoblasts is technically difficult due to their large cell size.

3. The authors state that Cre-negative (Cdh5fl/+ and Cdh5fl/fl) and Cre-positive heterozygous (Cyp19Cre; Cdh5fl/+) littermates were used as controls. Since Cre toxicity is a well-known problem, it should be shown which type of control mice were used in which experiments.

The controls in every experiment and analysis consists of embryos and placentas of both Cre-negative (Cdh5^fl/+^ and Cdh5^fl/fl^) and Cre-positive heterozygous (Cyp19^Cre^; Cdh5^fl/+^) genotypes. Cyp19^Cre^; Cdh5^fl/+^ mice are born at expected mendelian ratios, survive until adulthood, and do not exhibit any notable defects compared to Cdh5^fl/+^ and Cdh5^fl/fl^ mice. To further assess an effect of Cre expression on placentation we have compared the weights of Cdh5^fl/+^ and Cdh5^fl/fl^ vs. Cyp19^Cre^; Cdh5^fl/+^ E12.5 placentas and embryos. These studies revealed no differences (now shown as Figure 1—figure supplement 1C, D). Lastly, we have included a Source Data File containing exact quantification and statistical values for each graph in the figures along with specific genotypes, including a specific breakdown of Cre-negative and Cre-positive heterozygous controls.

Reviewer #2 (Recommendations for the authors):1) It should be demonstrated that Cyp19-Cre driven gene inactivation of VE-cadherin does not affect VE-cadherin expression in endothelial cells of the embryo in tissues where bleeding was observed.

The Cyp19-Cre is a very interesting Cre transgene. It was shown in the original paper describing these mice that the 501bp region within the first untranslated exon contains regulatory sequences specific for trophoblasts, whereas exons 2-10 encode the actual cytochrome P450 protein coding sequences that are expressed in liver, brain, gonads, etc^1^. The authors in this original paper specifically chose a clone in which Cre activity was present in trophoblasts and not in the embryo by LacZ reporter and X-gal staining, and multiple studies since then using fluorescent reporters also show no Cre activity in the embryo^2–4^.

Reviewer 2’s concerns regarding fetal hemorrhage due to embryonic loss of VE-cadherin were shared by Reviewer #3. Several lines of evidence support the conclusion that embryonic hemorrhage is secondary to placental defects rather than loss of VE-cadherin in embryo endothelial cells. First, staining of E12.5 Cyp19^Cre^; Cdh5^fl/fl^ E12.5 embryos for VE-cadherin shows that VE-cadherin levels are unaffected in endothelial cells within the embryo, including in the heart, liver, lungs, thorax, and at sites of hemorrhage in the brain. These new data are shown in Figure 1—figure supplement 3A-D. Second, we observed highly variable degrees of hemorrhaging in knockout embryos, consistent with a secondary event due to placental defects. To better demonstrate this, we have included additional images of an embryo and placenta in Figure 1. Finally, placental defects have been shown to confer secondary defects in vascular development^5^. We conclude that hemorrhage observed in knockout embryos likely results from global ischemia secondary to reduced maternal blood flow rather than loss of VE-cadherin within the embryonic vasculature. We have revised the results and discussion to better address the potential causes of embryonic hemorrhage.

2) The results shown in figures 2 and 3 should be confirmed by whole mount staining of either vibratome sections or cleared tissue.

This is an excellent suggestion by the reviewer. We performed whole-mount immunofluorescence of thick vibratome sections from control and Cdh5 knockout placentas. Optical sectioning and reconstruction with maximum intensity projections further support data from thin paraffin sections. In contrast to control placentas, Cdh5 knockout placentas maintain a layer of spiral artery endothelial cells despite being surrounded by trophoblasts and have decreased lumen size compared with control, trophoblast-lined spiral arteries. In the labyrinth, fewer maternal RBCs are observed in Cdh5 knockout placentas while the fetal vasculature is unaffected. These data are now included as Figure 2G, H. We thank the reviewer for these and other valuable suggestions.

3) It should be attempted to get some mechanistic insights by analyzing the effects of VE-cadherin silencing on the expression of genes that could be relevant for cell migration, collective migration or association of trophoblast cell clusters.

More detailed mechanistic insight obtained through analysis of gene expression was also requested by Reviewer #3, who suggested scRNA-seq to look at trophoblast gene expression. As discussed below, technical limitations precluded us from specifically isolating invasive trophoblasts for transcriptional analysis. We instead performed bulk RNA-seq on whole control and Cdh5 knockout deciduas. Analysis of the top 100 differentially regulated genes showed multiple genes related to decidual stromal dysfunction^6^. Additionally, GO analysis showed upregulation in pathways related to the ECM and immune cells in Cdh5 knockout placentas. These data are now included as Figure 4 in the manuscript.

Based on these RNA-seq data, we further investigated the membrane metalloprotease MMP15, which is associated with trophoblast invasion, and its ECM target laminin^7–9^. We also examined vinculin, a focal adhesion protein that promotes cell invasion. While we observed increased laminin in the knockout decidua, we found this was not due to a cell autonomous loss of MMP15. However, vinculin levels in trophoblasts were decreased which likely contributes to diminished trophoblast invasion. These data are now included as Figure 4—figure supplement 1. Based on GO analysis, we also evaluated uterine natural killer (uNK) cells in the decidua, which are the only innate immune cells found in the maternal decidual compartment. We found that Cdh5 knockout placentas had elevated uNK cells due to decreased apoptosis of those cells. As discussed in the revised manuscript, we believe that the uNK phenotype is secondary to failure of trophoblast migration into the decidua. These data are now included as Figure 4—figure supplement 2 in the manuscript. Together, these studies suggest that VE-cadherin contributes to focal adhesion formation in trophoblasts, enabling their migration and remodeling of the decidual ECM and immune cell microenvironment.

4) The discussion is rather short and does not attempt to debate mechanistic aspects of the findings.

In light of our new results, we have revised the discussion to incorporate more mechanistic aspects of our work, such as how VE-cadherin regulates vinculin expression in trophoblasts to promote invasion to trigger decidual ECM remodeling and immune changes that affects spiral artery remodeling. Additionally, we have incorporated discussion of potential strategies for promoting VE-cadherin expression in trophoblasts as suggested by Reviewer 3.

Reviewer #3 (Recommendations for the authors):The questions and suggestions below are therefore not intended to trigger significant additional experimental work for the present paper, but to perhaps increase clarity and make the study more accessible to readers.1. Please explain why the Cdh5 knockout embryos exhibited marked hemorrhage at E12.5 (Figure 1E).

As discussed above in response to Reviewer 2’s first comment, placental defects have been reported to cause secondary defects in vascular development^5^. In the revised manuscript we analyze VE-cadherin expression in control and knockout embryo vessels and demonstrate that hemorrhage and growth restriction are not due to loss of endothelial VE-cadherin within the embryo (Figure 1—figure supplement 3A-D). VE-cadherin is present in the vessels of all organs in control and knockout embryos, including the brain, heart, liver, lungs, and thorax. Thus, hemorrhage observed in knockout embryos likely results from tissue degradation associated with global ischemia secondary to reduced maternal blood flow rather than loss of VE-cadherin within the embryonic vasculature. Additionally, we wished to clarify that we observed highly variable degrees of hemorrhaging in knockout embryos. To better demonstrate this, we have included additional images of a knockout placenta and embryo in. We have revised the results and discussion to better explain potential causes of embryonic hemorrhage.

2. What about the diameter of the spiral artery in the Cdh5 knockout placentas versus WT placentas?

We thank Reviewer 3 for this excellent suggestion. Indeed, previous studies have shown that reduced remodeling of the spiral arteries and retention of vascular smooth muscle cells results decreased spiral artery lumen diameter^10^. Since the spiral arteries are morphologically tortuous, we measured diameters of spiral arteries only from those with circular cross sections (detailed in the Methods section). We found that Cdh5 knockout placentas had significantly smaller spiral artery diameters compared to controls, consistent with defects in spiral artery remodeling (revised Figure 2C).

3. It would be nice if the authors performed scRNA-seq of the trophoblasts of WT and Cdh5 knockout placentas and compared them to provide molecular insight into how the lack of Cdh5 in the trophoblasts led to such abnormalities.

We agree with the reviewer that single cell RNA-seq of placentas would be very insightful. Transcriptional analysis of trophoblasts at the single cell level in mice is technically challenging since trophoblast cells are extremely large and not amenable to single cell isolation. Some groups have overcome this with single nucleus RNA-seq; however this is also technically challenging since some trophoblasts are polyploid. We therefore performed bulk RNA-seq on whole deciduas from WT and Cdh5 knockout placentas. Bulk RNA-seq of whole deciduas from WT and Cdh5 knockout placentas revealed that several processes are disrupted in knockout placentas. Analysis of the top 100 differentially regulated genes showed multiple genes related to decidual stromal dysfunction^6^. Additionally, GO analysis showed upregulation in pathways related to the ECM and immune cells in Cdh5 knockout placentas. These data are included as a new Figure 4 in the manuscript.

Based on these RNA-seq data, we further investigated the membrane metalloprotease MMP15, which is associated with trophoblast invasion, and its ECM target laminin^7–9^. We also examined vinculin, a focal adhesion protein that promotes cell invasion. While we observed increased laminin in the decidua, we found this was not due to a cell autonomous loss of MMP15. However, vinculin levels in trophoblasts were decreased which likely contributes to diminished trophoblast invasion. These data are now included as Figure 4—figure supplement 1. Based on GO analysis, we also evaluated uterine natural killer (uNK) cells in the decidua, which are the only innate immune cells found in the maternal decidual compartment. We found that Cdh5 knockout placentas had elevated uNK cells due to decreased apoptosis. These data are now included as Figure 4—figure supplement 2 in the manuscript. Together, these studies suggest that loss VE-cadherin disrupts focal adhesions in trophoblasts, resulting in decreased migration and failure to remodel the decidual ECM and immune cell microenvironment.

4. Please describe any plausible ways to normalize Cdh5 level in the trophoblasts for enhanced endovascular invasion, which can be an alternative way for prevention of preeclampsia.

We agree that identification of strategies that promote Cdh5 expression in trophoblasts may be a valuable opportunity for treatment of preeclampsia. One study has shown that treatment of primary rat trophoblasts with VEGF-A induces VE-cadherin expression in vitro^11^. Thus, VEGF-A administration may be a useful therapy in promoting trophoblast invasion through upregulation of VE-cadherin. We have incorporated these potential strategies into our discussion. We thank Reviewer 3 for this and the other valuable suggestions.

References

1. Wenzel PL, Leone G. Expression of Cre recombinase in early diploid trophoblast cells of the mouse placenta. *genesis*. 2007;45(3):129-134. doi:10.1002/dvg.20276

2. López-Tello J, Pérez-García V, Khaira J, et al. Fetal and trophoblast PI3K p110α have distinct roles in regulating resource supply to the growing fetus in mice. *eLife*. 2019;8:1-25. doi:10.7554/*eLife*.45282.001

3. Sandovici I, Georgopoulou A, Pérez-García V, et al. The imprinted Igf2-Igf2r axis is critical for matching placental microvasculature expansion to fetal growth. *Dev Cell*. 2022;57(1):63-79.e8. doi:10.1016/j.devcel.2021.12.005

4. Vacher CM, Lacaille H, O’Reilly JJ, et al. Placental endocrine function shapes cerebellar development and social behavior. *Nat Neurosci*. 2021;24(10):1392-1401. doi:10.1038/s41593-021-00896-4

5. Perez-Garcia V, Fineberg E, Wilson R, et al. Placentation defects are highly prevalent in embryonic lethal mouse mutants. *Nature*. 2018;555(7697):463-468. doi:10.1038/nature26002

6. Woods L, Perez-Garcia V, Kieckbusch J, et al. Decidualisation and placentation defects are a major cause of age-related reproductive decline. *Nat Commun*. 2017;8(1):1-14. doi:10.1038/s41467-017-00308-x

7. Majali-Martinez A, Hoch D, Tam-Amersdorfer C, et al. Matrix metalloproteinase 15 plays a pivotal role in human first trimester cytotrophoblast invasion and is not altered by maternal obesity. *FASEB J*. 2020;34(8):10720-10730. doi:10.1096/fj.202000773R

8. Pollheimer J, Fock V, Knöfler M. Review: The ADAM metalloproteinases – Novel regulators of trophoblast invasion? *Placenta*. 2014;35(SUPPL):57-63. doi:10.1016/j.placenta.2013.10.012

9. Majali-Martinez A, Hiden U, Ghaffari-Tabrizi-Wizsy N, Lang U, Desoye G, Dieber-Rotheneder M. Placental membrane-type metalloproteinases (MT-MMPs): Key players in pregnancy. *Cell Adhes Migr*. 2016;10(1-2):136-146. doi:10.1080/19336918.2015.1110671

10. Burke SD, Barrette VF, Bianco J, et al. Spiral arterial remodeling is not essential for normal blood pressure regulation in pregnant mice. *Hypertension*. 2010;55(3):729-737. doi:10.1161/HYPERTENSIONAHA.109.144253

11. Chang CC, Chang TY, Yu CH, Tsai ML. Induction of VE-cadherin in rat placental trophoblasts by VEGF through a NO-dependent pathway. *Placenta*. 2005;26(2-3):234-241. doi:10.1016/j.placenta.2004.06.002